# A Multi-Agent Pipeline for Source-Grounded Synthetic Note Generation from Longitudinal Structured EHR

Nina Fatehi [1]   Reihaneh Hassanzadeh [1]   Meysam Ghaffari [1]   Animesh Agrawal [1]   Carlos Morato [1]

## Abstract

Structured EHR is abundant but sparse, coded, and difficult to use directly for note-centric clinical modeling. We present MedNotes, a multi-agent synthetic data generation pipeline that converts longitudinal structured EHR into source-grounded clinical note representations under explicit quality control. MedNotes treats structured-data-to-text synthesis as a closed-loop agentic process: a generator proposes a note, evaluator agents identify factual, coverage, structural, and hallucination-related failures, and an automatic routing component accepts, revises, or rejects the draft. On 1,485 EHRSHOT encounters, MedNotes achieves a 91.4% pass rate, with mean factual accuracy of 0.980, completeness of 99.1%, structural fidelity of 0.761, and 0.028 critical hallucinations per encounter. Iterative refinement improves acceptance from 69.4% to 91.4%. The resulting synthetic corpus improves downstream CPT prediction and paragraph-level section prediction when combined with limited real data.

## 1. Introduction

Structured EHR is abundant but difficult to use directly for many downstream clinical language-modeling tasks because it is sparse, coded, longitudinal, and often lacks shareable free-text notes. At the same time, authentic clinical notes are valuable for model development but difficult to release at scale because of privacy, governance, and annotation constraints. This has motivated growing interest in synthetic clinical text and source-grounded data generation, especially when synthetic artifacts preserve task-relevant structure while remaining traceable to the underlying data (Nadas et al., 2025; Lupidi et al., 2024; Amad et al., 2025). Recent work on tabular generation and structured-data seri-

alization suggests that language-model-style representations can serve as useful interfaces for structured records (Borisov et al., 2023; Hegselmann et al., 2023; Solatorio & Dupriez, 2023; Hollmann et al., 2023). Most clinical note generation work assumes richer source modalities, such as encounter dialogue, audio, or paired clinician-authored notes, as in MEDIQA-Chat and ACI-Bench (Abacha et al., 2023; Yim et al., 2023). In many realistic settings, however, the available data are longitudinal structured EHR: diagnoses, medications, procedures, laboratory results, vital signs, and coded observations. EHRSHOT is a representative example of this structured-only setting (Wornow et al., 2023). This creates a modality gap: many downstream note-centric models expect text, while the most accessible clinical data source is often coded EHR.

We study this setting as a quality-gated synthetic data generation problem. The goal is not to reproduce clinician-authored documentation, but to construct source-grounded clinical note representations that can support downstream adaptation, benchmarking, and structured-data learning. We instantiate the target format as SOAP because its sectioned anatomy provides a useful interface for controlled generation: it separates evidence-bearing content from interpretation and planning, supports section-aware feedback, and enables structural consistency checks.

We introduce MedNotes, a multi-agent pipeline for converting longitudinal structured EHR into source-grounded synthetic clinical notes. MedNotes treats structured-data-to-text synthesis as a closed-loop agentic process: a generator proposes a note, evaluator agents identify factual, coverage, structural, and hallucination-related failures, an aggregator constructs repair signals and preservation anchors, and a router accepts, revises, or rejects the draft under explicit quality constraints. This adapts iterative feedback and textual-gradient ideas (Madaan et al., 2023; Shinn et al., 2023; Pryzant et al., 2023) to quality-gated synthetic corpus construction from structured clinical data.

## 2. Methods and Cohort

We present MedNotes, a controlled synthetic data generation pipeline that transforms longitudinal structured EHR into source-grounded clinical note representations under ex-

---

[1] Optum AI, UnitedHealth Group, Minneapolis, Minnesota, USA. Correspondence to: Carlos Morato <carlos.morato@optum.com>.

*Proceedings of the 2nd ICML Workshop on Foundation Models for Structured Data*, Seoul, South Korea. 2026. Copyright 2026 by the author(s).

plicit quality gating. We instantiate the target representation as SOAP because its sectioned structure supports modular prompting, feedback assignment, and structural evaluation, but the method is intended more broadly as quality-gated structured-data-to-text synthesis.

## 2.1. Task Formulation

Let $e_i$ denote the current encounter for patient $p$ at index $i$, and let $H_i^{(k)}$ denote up to $k = 2$ prior encounters used as longitudinal context. The generator input is

$$x_i = \left( e_i, H_i^{(k)} \right), \qquad (1)$$

where encounters contain structured clinical information such as diagnoses, medications, procedures, laboratories, and vital signs after code resolution into human-readable text. Given $x_i$, the system generates a synthetic sectioned clinical note $n_i$.

Accepted outputs form a synthetic corpus

$$\mathcal{D}_{\mathrm{syn}} = \{(x_i, n_i) : \mathrm{PASS}(n_i, x_i) = 1\}, \qquad (2)$$

intended for downstream adaptation, benchmarking, and representation learning when authentic notes or paired supervision are unavailable. Because structured EHR rarely contains recoverable patient-reported narrative, the Subjective section is treated conservatively: the model may summarize supported contextual information but must abstain from inventing unsupported patient-reported details.

For each candidate note, MedNotes evaluates four quality dimensions:

$$
\begin{aligned}
f_i &= \mathrm{Fact}(n_i, x_i) \in [0, 1], \\
c_i &= \mathrm{Comp}(n_i, x_i) \in [0, 100], \\
s_i &= \mathrm{SFS}(n_i, x_i) \in [0, 1], \\
h_i^{\mathrm{crit}} &= \mathrm{CritHall}(n_i, x_i) \in \mathbb{N}_0,
\end{aligned} \qquad (3)
$$

where $f_i$ is factual accuracy, $c_i$ is entity completeness, $s_i$ is structural fidelity Score (SFS) under the target note format, and $h_i^{\mathrm{crit}}$ is the number of critical hallucinations. A note is accepted only if

$$
\begin{aligned}
\mathrm{PASS}(n_i, x_i) = \mathbb{I}\big[ &f_i \geq 0.95 \ \wedge \ c_i \geq 90 \\
&\wedge \ s_i \geq 0.67 \ \wedge \ h_i^{\mathrm{crit}} = 0 \big].
\end{aligned} \qquad (4)
$$

Generation is iterative. Let $n_i^{(0)}$ be the initial draft. After round $r$, evaluator agents produce section-specific corrective feedback $\Gamma_i^{(r)}$ and positive anchors $A_i^{(r)}$ for already-correct content:

$$
\begin{aligned}
n_i^{(0)} &= G(x_i), \\
n_i^{(r+1)} &= G\left( x_i, n_i^{(r)}, \Gamma_i^{(r)}, A_i^{(r)} \right).
\end{aligned} \qquad (5)
$$

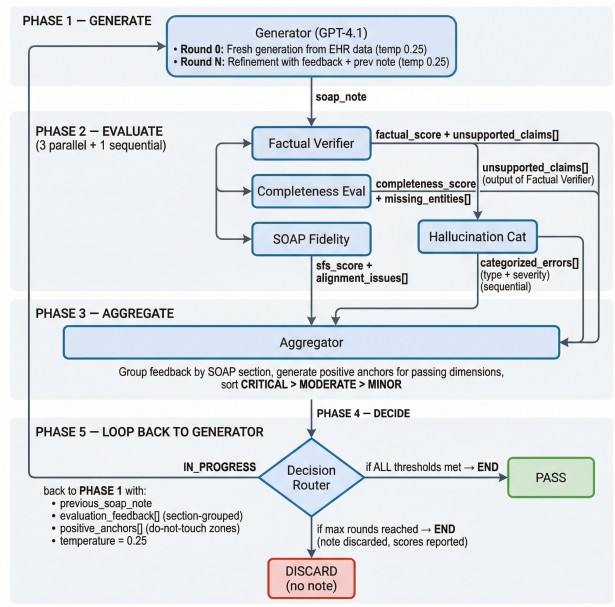

*Figure 1.* MedNotes pipeline architecture. Each encounter passes through generation, evaluation, aggregation, and decision stages. Factual verification, entity completeness, and structural fidelity are computed in parallel, followed by hallucination categorization. Evaluator outputs are converted into section-specific repair signals and positive anchors for the next round.

The pipeline runs for at most $R_{\max} = 3$ rounds. If no draft satisfies the acceptance criterion, the system abstains and returns no note.

## 2.2. Dataset and Cohort

We evaluate on 100 patients from EHRSHOT, a de-identified longitudinal EHR benchmark spanning 2009–2022 (Wornow et al., 2023). Patients were selected by balanced stratified sampling across diabetes, hypertension, obesity, and a general cohort. The final evaluation set contains 1,485 encounters, with 10–20 encounters per patient. Each encounter includes structured codes from multiple vocabularies, including ICD-10-CM, RxNorm, CPT, LOINC, and SNOMED. All codes are resolved to human-readable descriptions before generation to reduce ambiguity and code-interpretation errors.

## 2.3. MedNotes Pipeline

MedNotes treats structured-EHR-to-text synthesis as a closed-loop agentic control problem rather than a one-shot prompting task. The generator proposes an initial note from the current encounter and limited longitudinal context. Evaluator agents then decompose note quality into complementary dimensions: factual support against the source EHR, recall-style coverage of structured entities, structural consistency across note sections, and severity of unsupported claims. The aggregator compiles evaluator outputs into

section-specific repair signals and positive anchors, and the automatic routing component either accepts the note, sends it back for revision, or rejects the encounter after the round budget is exhausted.

The generator is GPT-4.1 (Azure OpenAI; temperature 0.25) with a conservative rule-based system prompt. In the first round, it generates a note from the current encounter and up to two prior encounters. In later rounds, it is re-invoked with the previous note, section-grouped feedback, and positive anchors. The prompt emphasizes resolved clinical descriptions rather than raw codes, comprehensive use of current-encounter evidence, qualified historical references, conservative handling of missing patient-reported content, and abstention from unsupported inference.

The evaluation layer uses specialized agents to measure source-grounded quality. A factual consistency evaluator compares note claims against structured EHR data.; a completeness evaluator measures coverage of clinically significant entities; a structural fidelity evaluator scores cross-section consistency; and a hallucination categorizer identifies and categorizes unsupported claims by type and severity. These outputs are merged into targeted feedback for the next generation round. A note is retained in $\mathcal{D}_{\mathrm{syn}}$ only when all thresholds are satisfied jointly; otherwise, the system revises the note or abstains after three rounds.

# 3. Experiments and Results

We evaluate three questions. First, can MedNotes construct source-grounded synthetic notes from structured longitudinal EHR under explicit quality gating? Second, how much does iterative evaluator-guided refinement improve acceptance? Third, do the accepted synthetic notes carry useful signal for downstream note-centric prediction tasks? The primary quality metric is encounter-level acceptance under Eq. 4; we also report final-round factual accuracy, entity completeness, structural fidelity, critical hallucinations, and total categorized error burden. Supporting development studies, including prompt optimization, lookback-window sensitivity, relevance-based history selection, the full error-recovery breakdown, and further downstream task details, are provided in the appendix.

## 3.1. Quality-Gated Synthesis on EHRSHOT

Table 1 summarizes aggregate system performance. MedNotes achieves a 91.4% encounter-level pass rate, with 1,357 accepted encounters and 128 discarded encounters. No API failures occurred during evaluation. Across final outputs, the system achieves a mean factual accuracy of 0.980, mean completeness of 99.1%, mean structural fidelity of 0.761, and 0.028 critical hallucinations per encounter. These results indicate that the multi-agent control loop can satisfy

strict source-grounded quality criteria for most encounters while preserving an explicit abstention mechanism for cases that remain too sparse or ambiguous. In a runtime benchmark over the same 1,485 encounters, MedNotes processed encounters in 76.8 seconds on average at an estimated API cost of $0.0958 per encounter (Appendix A); this reflects the deployed API configuration for this run, and model versions and pricing may differ across configurations.

*Table 1.* Overall results on the 100-patient evaluation cohort.

| Metric | Value |
| --- | --- |
| Total encounters | 1,485 |
| Pass | 1,357 (91.4%) |
| Discarded | 128 (8.6%) |
| API failures | 0 (0.0%) |
| Mean factual accuracy (final) | 0.980 |
| Mean completeness (final) | 99.1% |
| Mean SFS (final) | 0.761 |
| Mean critical hallucinations / encounter | 0.028 |
| Mean total categorized error burden / encounter | 0.344 |

## 3.2. Reflection and Error Recovery

To isolate the contribution of iterative reflection, we analyzed the round at which each encounter first satisfied the joint acceptance criterion. As shown in Table 2, 1,031 encounters passed in Round 1, corresponding to 69.4% of all encounters and 76.0% of accepted encounters. Round 2 added 266 accepted encounters and Round 3 added 60 more, yielding a final pass rate of 91.4%.

Without reflection, MedNotes would achieve only a 69.4% pass rate. The iterative repair loop therefore contributes a net gain of 22.0 percentage points. Among the 454 encounters that failed in Round 1, reflection recovered 326, corresponding to a 71.8% recovery rate. A finer-grained analysis showed that isolated hallucination and factuality failures were more recoverable than persistent structural or completeness failures. Full recovery breakdowns are in Appendix C.

*Table 2.* Pass rate by reflection round.

| Round | Pass | % of accepted | Cumulative |
| --- | --- | --- | --- |
| Round 1 | 1,031 | 76.0% | 69.4% |
| Round 2 | 266 | 19.6% | 87.3% |
| Round 3 | 60 | 4.4% | 91.4% |
| Discarded | 128 | — | — |

## 3.3. Preliminary Clinician Review

We obtained exploratory feedback from a practicing physician on a small subset of generated notes, focusing on source support, clinically important omissions, and overall usefulness. This was intended as a qualitative expert assessment rather than a formal human-evaluation study.

Across reviewed cases, the physician feedback was consistent with the intended conservative operating point of MedNotes. Accepted notes were readable and source-grounded, but often reflected the sparsity of the underlying structured EHR: they summarized coded events and available context rather than adding unsupported narrative detail, exam findings, or clinical reasoning absent from the source. This supports our framing of the current pass criterion as a source-grounded quality gate for synthetic corpus construction, rather than a guarantee of clinician-level documentation utility.

### 3.4. Downstream Task Utility

We next tested whether the accepted synthetic corpus carries reusable note-centric signal in downstream prediction tasks. We evaluate two settings: CPT prediction from notes and paragraph-level section prediction.

#### 3.4.1. CPT PREDICTION FROM SYNTHETIC NOTES

Given a clinical note, the model predicts the set of CPT codes associated with the encounter. This is formulated as a multi-label prediction problem. We compare an untuned Llama-3 8B baseline, fine-tuning on 100 MIMIC note–CPT pairs, and a mixed-data setting using synthetic notes plus 100 MIMIC examples. Table 3 reports performance on a held-out MIMIC test set (Johnson et al., 2016). The mixed synthetic+MIMIC setting achieved the best performance across all reported metrics, suggesting that synthetic notes provide useful task-relevant supervision when combined with limited real data.

Table 3. CPT prediction results on the held-out MIMIC test set.

| Model | Avg. | P | R | F1 |
|---|---|---|---|---|
| Base Llama-3 8B Instruct | Macro | 0.0152 | 0.0634 | 0.0234 |
| | Micro | 0.0366 | 0.0688 | 0.0478 |
| + Fine-tuned on 100 MIMIC | Macro | 0.0173 | 0.0665 | 0.0244 |
| | Micro | 0.0441 | 0.0516 | 0.0476 |
| + Synthetic + 100 MIMIC | Macro | 0.2137 | 0.1034 | 0.1030 |
| | Micro | 0.0827 | 0.1204 | 0.0980 |

#### 3.4.2. PARAGRAPH-LEVEL SECTION PREDICTION

We also evaluate paragraph-level section prediction, where each paragraph is classified as Subjective, Objective, Assessment, or Plan. Using MediSOAP as the evaluation benchmark (MediSOAP Contributors, 2023), we compare training on 100 real notes against training on synthetic data combined with 100 real notes. As shown in Table 4, the combined model reaches 0.972 accuracy and 0.972 macro-F1, achieving higher results on 100 real notes alone (see Appendix F for section-wise performance). This indicates that the synthetic corpus provides useful complementary supervision for structural note understanding.

Table 4. Overall performance on the MediSOAP test set.

| Model | Accuracy | Macro-F1 |
|---|---|---|
| BERT fine-tuned on 100 MediSOAP notes | 0.785 | 0.752 |
| BERT fine-tuned on synthetic + 100 MediSOAP notes | **0.972** | **0.972** |

## 4. Discussion

MedNotes is a quality-gated synthetic data generation framework for converting sparse, coded, longitudinal EHR into source-grounded text representations. Rather than introducing a new foundation model, its main contribution is a closed-loop agentic controller for structured-data-to-text synthesis: evaluator agents identify unsupported claims, missing entities, structural inconsistencies, and hallucination risks; the aggregator converts these signals into repair instructions and preservation anchors; and the automatic routing component accepts, revises, or discards drafts. This makes synthetic corpus construction more controlled than one-shot prompting when generated text may later be used for downstream model training.

The results suggest that accepted synthetic notes can serve as useful intermediate representations when authentic notes are unavailable, especially when combined with limited real data. At the same time, preliminary clinician review clarifies the operating point: source-groundedness and structural consistency do not guarantee clinical richness, particularly when structured EHR lacks detailed subjective narrative, exam findings, or clinical reasoning. Future work should improve salience-aware history selection and synthesis while preserving the evidence constraints that make the corpus traceable.

## 5. Conclusion

We presented MedNotes, a multi-agent pipeline for quality-gated synthetic note generation from longitudinal structured EHR. The system couples generation, evaluator-guided refinement, and abstention to construct source-grounded clinical text representations from sparse coded records. On EHRSHOT, MedNotes achieves high acceptance under strict quality thresholds, and downstream experiments show that the resulting synthetic corpus provides useful signal for clinical code prediction and section prediction when paired with limited real data. Taken together these findings may support quality-gated synthetic notes as a practical bridge between structured EHR and note-centric machine learning, while motivating future work on richer clinical salience and context selection.

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

# A. Cost and Latency Analysis

We measured runtime and API cost over the 1,485-encounter evaluation run. Cost estimates use recorded token usage from the deployed API configuration and the model prices at the time of execution. These costs exclude engineering, storage, orchestration overhead, and human review time.

*Table 5.* System cost and latency analysis over 1,485 encounters. Token counts are empirical estimates and may vary with encounter complexity.

| Metric | Total | Per encounter |
|---|---|---|
| *Latency* | | |
| Processing time | – | 76.8s |
| Throughput | – | 46.9 enc/hr |
| *Rounds executed* | | |
| Total rounds | 2,087 | 1.41 |
| Terminal after Round 1 | 1,056 | 71.1% |
| Terminal after Rounds 2–3 | 429 | 28.9% |
| *Token usage* | | |
| Generation input tokens | 7.30M | 4,919 |
| Generation output tokens | 1.88M | 1,265 |
| Evaluation input tokens | 16.70M | 11,243 |
| Evaluation output tokens | 4.17M | 2,811 |
| *API cost (USD)* | | |
| Generation model | $29.64 | $0.0200 |
| Evaluation models | $112.70 | $0.0759 |
| **Total** | **$142.33** | **$0.0958** |

# B. Generator Prompt Rule Summary

The generator prompt enforces a conservative source-grounded operating point. The main rules are: (1) use resolved clinical descriptions rather than raw codes; (2) include all current-encounter structured evidence; (3) follow strict section structure; (4) use prior encounters only as contextual support; (5) qualify all historical references; (6) do not infer absent findings; (7) treat Subjective content as a conservative proxy when patient-reported narrative is unavailable; (8) handle medication continuity conservatively and never infer start dates; (9) keep Objective evidence-bearing only; (10) restrict Assessment to current-encounter-grounded impressions; and (11) keep Plan grounded and non-empty. Together, these rules prioritize traceability and hallucination control over unconstrained narrative richness.

# C. Error Analysis of Round-1 Failures

Table 6 reports the full recovery analysis for the 454 encounters that failed in Round 1. Categories are mutually exclusive and defined from Round-1 threshold violations only. Recovery denotes eventual pass after up to two additional refinement rounds.

*Table 6.* Error trajectory of Round-1 failures. Confidence intervals are patient-clustered 95% bootstrap intervals.

| Failure mode at Round 1 | $N$ | Recovered | Recovery rate (95% CI) | Discard rate |
|---|---|---|---|---|
| Critical hallucination | 39 | 34 | 87.2% [76.9, 97.1] | 12.8% |
| Factuality failure | 158 | 124 | 78.5% [70.9, 86.0] | 21.5% |
| SFS failure | 116 | 71 | 61.2% [51.1, 70.8] | 38.8% |
| Completeness failure | 16 | 8 | 50.0% [25.0, 75.0] | 50.0% |
| Multiple simultaneous | 125 | 89 | 71.2% [63.1, 79.1] | 28.8% |
| Total | 454 | 326 | 71.8% [67.2, 76.4] | 28.2% |

# D. Prompt Optimization on the Development Cohort

We performed offline prompt optimization on a fixed 10-patient development cohort containing 155 encounters. Each cycle followed three phases: diagnose recurring failures, modify the base prompt, and validate on the same cohort. Table 7 summarizes the version progression.

*Table 7.* Prompt version comparison on the 10-patient development cohort.

| Version | Pass | Disc. | Rate | R0 Pass | R1 Rescue | R2 Rescue |
|---------|------|-------|------|---------|-----------|-----------|
| v1 | 137 | 18 | 88.4% | 89 (57.4%) | 37 (23.9%) | 11 (7.1%) |
| v2 | 130 | 25 | 83.9% | 70 (45.2%) | 45 (29.0%) | 15 (9.7%) |
| v3 | 142 | 13 | 91.6% | 101 (65.2%) | 27 (17.4%) | 14 (9.0%) |
| v4 | 146 | 9 | 94.2% | 94 (60.6%) | 37 (23.9%) | 15 (9.7%) |

## E. History-Selection Sensitivity Studies

We conducted two supporting analyses for longitudinal context. First, a pre-agentic lookback-window study varied $k \in \{0, 1, 2, 5\}$ using the earlier single-shot generator. Completeness increased only modestly from $k = 0$ to $k = 5$, while available factuality runs suggested possible carry-forward risk at longer windows; this motivated the default choice of $k = 2$. Second, a 30-patient relevance-based history pilot compared the default recent-2 policy against selecting the two most similar prior encounters under a structured overlap heuristic. The relevance-based selector underperformed the recent-2 baseline (60% vs. 87% pass rate), so the final system retained the recent-history policy.

*Table 8.* Pilot comparison of recent-history and relevance-based history selection.

| Metric | Recent-2 | Relevant-2 |
|--------|----------|------------|
| Pass rate | 26/30 (87%) | 18/30 (60%) |
| Mean factual accuracy | 98.28% | 85.67% |
| Mean completeness | 97.92% | 82.69% |
| Mean structural fidelity | 74.72% | 75.19% |

## F. Detailed Section Prediction Results

Table 9 reports section-wise precision, recall, and F1 for the paragraph-level section prediction task.

*Table 9.* Section-wise performance on the MediSOAP test set.

| Model | Section | Precision | Recall | F1 |
|-------|---------|-----------|--------|-----|
| MediSOAP-100 | Subjective | 1.000 | 0.228 | 0.371 |
| MediSOAP-100 | Objective | 0.553 | 0.976 | 0.706 |
| MediSOAP-100 | Assessment | 0.976 | 0.968 | 0.972 |
| MediSOAP-100 | Plan | 0.953 | 0.968 | 0.960 |
| Synthetic | Subjective | 0.887 | 0.284 | 0.430 |
| Synthetic | Objective | 0.946 | 0.348 | 0.509 |
| Synthetic | Assessment | 0.485 | 0.976 | 0.648 |
| Synthetic | Plan | 0.769 | 1.000 | 0.870 |
| Synthetic + MediSOAP-100 | Subjective | 0.939 | 0.988 | 0.963 |
| Synthetic + MediSOAP-100 | Objective | 0.983 | 0.916 | 0.948 |
| Synthetic + MediSOAP-100 | Assessment | 0.992 | 0.984 | 0.988 |
| Synthetic + MediSOAP-100 | Plan | 0.977 | 1.000 | 0.988 |

