# OpenReview forum: "A Multi-Agent Pipeline for Source-Grounded Synthetic Note Generation from Longitudinal Structured EHR"
_ICML.cc/2026/Workshop/FMSD — FMSD @ ICML 2026 Poster_

### Official Review · Reviewer_zcag · 2026-05-16
**a multi-agent pipeline for converting longitudinal structured EHR encounters into source-grounded SOAP-style synthetic notes**

**Rating:** 6
**Confidence:** 3

**Review:**

Strengths:

 - Reporting cost, latency, token use, number of rounds, and no API failures over the full 1,485-encounter run gives readers concrete evidence about the practical overhead of the pipeline. These details make the work easier to assess as an applied synthetic-data system.
 - Conservative treatment of the Subjective section is appropriate for a structured-only EHR source. The paper correctly avoids inventing patient-reported narrative when the source data mainly contains diagnoses, medications, procedures, labs, vitals, and codes.

Cons:

 - central quality metrics are not independently validated. Factual accuracy, completeness, structural fidelity, and critical hallucinations appear to be produced by automated evaluator agents that also drive revision and acceptance, but the paper does not provide evaluator prompts, model identities, calibration data, agreement with human annotations, or error bars for the main final metrics. As written, the strongest result is that the system learns to pass its own gate, not that the accepted notes are verified source-grounded clinical artifacts.
 - clinician review is too thin to support the quality claims. The manuscript says a practicing physician reviewed a small subset, but does not report the sample size, sampling method, rubric, blinded status, agreement with automated decisions, or examples of missed errors. A blinded review of both accepted and rejected notes would be much more informative.
 - CPT prediction experiment has a serious leakage risk because the generation input includes CPT and procedure information, while the downstream model is asked to predict CPT codes from the generated note. If label-derived procedure descriptions appear in the synthetic notes, the gain may reflect direct label leakage rather than clinically meaningful note understanding.
 - downstream tasks are confounded by domain and format. Synthetic notes are generated from EHRSHOT but CPT prediction is tested on MIMIC, and section prediction uses SOAP-formatted synthetic notes to classify SOAP sections in MediSOAP. The section task in particular needs header-stripped inputs, ordering-cue controls, and a rule-based baseline to show that performance is not mainly driven by template cues.

---

### Official Review · Reviewer_H9SG · 2026-05-20
**Source Grounded Synthetic Note Generation**

**Rating:** 6
**Confidence:** 4

**Review:**

Summary: The paper proposes a framework for transforming structured EHR codes into synthetic SOAP notes using a generate-evaluate-revise loop. A generator LLM drafts notes from coded encounters, four evaluator agents score factual accuracy, completeness, structure, and hallucinations, and a router decides whether to accept, revise, or discard. On 1,485 EHRSHOT encounters, 91.4% pass a joint quality gate. Iterative refinement adds 22 points over single-shot generation. Downstream experiments on CPT prediction and section classification show gains when synthetic notes supplement small amounts of real data.


Strengths: The paper proposes a rationale framework decomposing evaluation into independent dimensions and feeding structured repair signals back to the generator avoids the usual problem of iterative refinement breaking things that were already correct. The most interesting finding in the paper is the error-recovery analysis (Table 6): it shows that hallucination failures recover at 87% while completeness failures recover at only 50%, which tells you something real about what LLM revision can and can't fix.


Weakness: The paper heavily relies on LLM as a judge. It would bring lot more credibility to the results if human evaluation results (even on a subset were reported). This is to understand how reliable LLM has been at judgement. The downstream CPT experiment adds more samples for synthetic+MIMIC condition, so it is hard to tell if the performance gains are from improved quality of data or just increase in the quantity. While it shows improvement, absolute F1 values are still below 0.1. No comparison against standard baselines.

---

### Official Review · Reviewer_7cG9 · 2026-05-20
**Review for "A Multi-Agent Pipeline for Source-Grounded Synthetic Note Generation from Longitudinal Structured EHR"**

**Rating:** 7
**Confidence:** 3

**Review:**

# Summary

This paper introduces a multi-agent pipeline MedNotes that converts longitudinal structured EHR into source-grounded synthetic SOAP notes under explicit quality gating. A generator drafts a note, parallel evaluators score factual accuracy, completeness, structural fidelity, and critical hallucinations, and a router accepts, revises, or discards the draft over up to three rounds. On 1,485 EHRSHOT encounters, MedNotes achieves a 91.4% pass rate, with iterative refinement contributing 22 percentage points over single-shot generation. The accepted synthetic corpus substantially improves downstream CPT code prediction and paragraph-level section classification when combined with limited real data, demonstrating practical value as a bridge between structured EHR and note-centric modeling.

# Strengths

- The paper addresses a clear, underexplored gap between abundant structured EHR and note-centric modeling.
- Its quality decomposition into four complementary dimensions, with joint acceptance and zero-tolerance for critical hallucinations, is principled.
- The closed-loop design with positive anchors yields a substantial 22-point pass-rate gain over single-shot generation.
- Downstream validation on real MIMIC and MediSOAP test sets convincingly demonstrates reusable signals.
- The abstention mechanism reflects responsible synthetic data construction.

# Areas for Improvement

- The paper does not describe how the evaluator agents produce continuous numerical scores. It is unclear whether the evaluators output ratings directly (e.g., 'this note's factual accuracy is 0.95') or whether scores are computed from structured intermediate products. Direct LLM scoring is known to suffer from bias and limited self-consistency, so the meaning of reported precision depends critically on this implementation choice.
- The experiment did not include comparison against non-agentic alternatives.
- The 91.4% pass rate measures whether MedNotes' own LLM evaluators accept the outputs. Three risks compound this: 1) if evaluators share a model family with the generator, they likely share its blind spots and stylistic preferences, producing systematically inflated scores; 2) LLM evaluators cannot recognize their own knowledge gaps in clinical reasoning, drug interactions, etc. Also, the preliminary clinician review is qualitative, does not report sample size, does not score the four dimensions, and provides no agreement assessments with the LLM evaluators, so the validation is a bit weak.
- Evaluation thresholds and the three-round budget are asserted without explicit justifications.

# Detailed Comments

On methodology, it is highly recommended that the paper clarify whether the four scores are emitted directly by the LLM evaluators or computed from structured intermediate outputs, since the meaning of the reported precision depends on this choice; relatedly, the evaluator prompts should be included in the appendix, along with the model type, version, and configuration of each evaluator agent, which are currently unspecified yet needed for reproducibility. Self-consistency tests, reporting variance under repeated evaluation of identical inputs, would help ground the precision of the reported metrics, and clarifying whether the evaluators share a model family with the generator, then replicating with a different family on a subset, would directly address the shared-bias concern that the headline pass rate rests on. A formal evaluator-clinician agreement study would also substantially strengthen the validation, specifying sample size, sampling strategy, and per-dimension scoring, ideally with inter-rater agreement against the LLM evaluators. Finally, a simpler non-agentic baseline would help isolate the multi-agent contribution, which is especially informative given the workshop's structured-data focus, and sensitivity analyses for the evaluator thresholds and the three-round budget would considerably strengthen the robustness claims.

# Justification of Score

Overall, this is a great work that addresses a clear gap between structured EHR and note modeling with a principled multi-dimensional quality framework, a well designed closed loop architecture with substantial pass-rate, and convincing downstream validation on real test sets.  However, the evaluator implementation is a bit opaque, and some of the experiments could be better established.